# Analysis and Optimization of a Microchannel Heat Sink with V-Ribs Using Nanofluids for Micro Solar Cells

**DOI:** 10.3390/mi10090620

**Published:** 2019-09-17

**Authors:** Ruijin Wang, Jiawei Wang, Weijia Yuan

**Affiliations:** 1School of Mechanical Engineering, Hangzhou Dianzi University, Hangzhou 310018, China; 2Bharti School of Engineering, Laurentian University, Sudbury, ON P3E 2C6, Canada

**Keywords:** micro solar cell, nanofluid, microchannel heat sink, V-ribs, heat transfer enhancement

## Abstract

It is crucial to control the temperature of solar cells for enhancing efficiency with the increasing power intensity of multiple photovoltaic systems. In order to improve the heat transfer efficiency, a microchannel heat sink (MCHS) with V-ribs using a water-based nanofluid as a coolant for micro solar cells was designed. Numerical simulations were carried out to investigate the flows and heat transfers in the MCHS when the Reynolds number ranges from 200 to 1000. The numerical results showed that the periodically arranged V-ribs can interrupt the thermal boundary, induce chaotic convection, increase heat transfer area, and subsequently improve the heat transfer performance of a MCHS. In addition, the preferential values of the geometric parameters of V-ribs and the physical parameters of the nanofluid were obtained on the basis of the Nusselt numbers at identical pump power. For MCHS with V-ribs on both the top and bottom wall, preferential values of V-rib are rib width d/W=1, flare angle α=75°, rib height hr/H=0.3, and ratio of two slant sides b/a=0.75, respectively. This can provide sound foundations for the design of a MCHS in micro solar cells.

## 1. Introduction

With the rapid increase of concentrated multiples in the photovoltaic system, it is crucial to improve the efficiency of micro solar cells by controlling cell temperature. An effective approach to decrease the temperature of a photovoltaic module is to equip a specialized cooling system [1,2,3,4,5]. The most common approaches to cool the solar cell are jet-impact cooling [6] and microchannel heat sinks (MCHSs) [7,8,9]. The MCHS was proposed initiatively in 1980s by Tuckeman [10] to replace the traditional heat exchanger for satisfying cooling requirements. Thereafter, a variety of approaches to elevate the heat transfer performance of MCHS were proposed, such as various shapes of microchannels [11,12], complex manifold geometries [13,14], double/multi-layer MCHSs [15,16,17] and interrupted microchannels [18,19]. Sidik [20] presented an overview of passive techniques for heat transfer augmentation in MCHS. The most important passive techniques are to interrupt the thermal boundary and to induce chaotic convection by various ribs (baffles) or/and ruts (grooves) periodically located on the wall. For example, Wang [21] performed a parametric investigation on the MCHS with slant rectangular ribs. Geometric parameters of slant rectangular ribs are optimized by evaluating the average Nusselt number at identical pump power. Behnampour [22] studied the heat transfer enhancement of MCHS with triangular, rectangular and trapezoidal ribs. The numerical results showed that the triangular ribs have the highest criteria value of thermal performance evaluation. Cai [23] conducted a numerical investigation on the heat transfer in MCHS with different shapes of offset ribs on sidewalls and indicated that the offset ribs result in significant heat transfer enhancement. Li [24] presented a novel MCHS with triangular cavities and rectangular ribs. The effects of cavities and ribs on the heat transfer performance of MCHS are investigated. Duangthongsuk [25] presented a comparison of the convective heat transfer, pressure drop, and performance index characteristics of heat sinks with a miniature circular pin—fin inline arrangement and a zigzag flow channel with single cross-cut structures. Wang [26] proposed a new double-layered microchannel heat sink with porous fins instead of solid fins. The new design yields a 45.3%–48.5% reduction in pumping power attributed to the “slip effect” of coolant on the porous fin walls. The effect of L-shaped porous ribs on the heat transfer of nanofluid flow inside a microchannel were studied by Toghraie [27]. The results indicated that the heat transfer rate when using porous ribs increases up to 42% and 25% at *Re* = 1200 and 100, respectively. Akbari [28] investigated the effect of a semi-attached rib on heat transfer and liquid turbulence of nanofluid in a three-dimensional (3D) rectangular microchannel. The effects of geometric parameters such as the dimensions of the semi-attached rib, the volume fraction of the nanoparticle, and Reynolds number were considered. Wang [29] proposed a high-performance MCHS with bidirectional ribs composed of a vertical rib (VR) and a spanwise rib (SR), because the bidirectional ribs can interrupt the thermal boundary layer and induce circulation in both vertical and spanwise directions.

Besides, combination of various structures can be employed to promote the advantages and weaken the disadvantages of a single structure. Ghani [30] proposed a novel MCHS with sinusoidal Cross–Mark cavities and rectangular ribs. They indicated that the thermal performance is superior owing to the combination of two important features: a significant reduction of the pressure drop, and violent flow turbulence. Interestingly, a new scheme wavy MCHS with wavy porous fins was proposed by Lu [31] for reducing pressure drop and thermal resistance simultaneously. The remarkable reduction of pressure drop comes from the combination of the permeation effect and the slip effect. The heat transfer enhancement is attributed to the combination of the mixing of fluid by Dean vortex and the forced permeation by jet-like impingement. Vinoth [32] experimentally investigated the heat transfer and flow characteristics of oblique finned MCHS and indicated that a trapezoidal cross-section performs better than square and semi-circle cross sections. Ghani [33] reviewed many published results and concluded that flow disruption for enhancing flow mixing, interruption of the boundary layer, and increasing heat transfer area are three important mechanisms to improve the performance of MCHS in passive structures. Sidik [20] summarized more comprehensively that impact factors, including channel curvature, surface roughness, flow disruption, out-of-plane mixing, secondary flow, reentrant obstruction and fluid additives, can improve the heat transfer performance. In general, the underlying mechanisms to enhance the heat transfer performance of MCHS include improving the thermal conductivity of the working medium, with the exception of the three fundamental mechanisms mentioned in [33]. 

Moreover, the use of nanofluid as a coolant can further promote the heat transfer performance of MCHS because the added nanoparticles can improve the thermal conductivity of the cooling medium owing to the Brownian motion and thermophoretic movement of nanoparticles. On the basis of micromechanical analysis, Buongiorno [34] concluded that the most important slip mechanisms between nanoparticles and base fluids are Brownian motion and thermophoresis. Consequentially, a two-component and four-equation model for nanofluids was established. Many researchers used this model to simulate heat transfer and flow in the MCHS using nanofluids [8,9,16,18,22,25,27]. Besides, Li [35] proposed a temperature decomposition method to simulate the heat transfer through periodic structures with different wall temperatures. Minea [36] numerically investigated the convective heat transfer characteristics using an Al_2_O_3_–water nanofluid and indicated the heat transfer coefficient increases by 3.4%–27.8%. Guo [37] numerically investigated the thermal and hydraulic performance in a micro fin heat sink (MFHS) with ZnO–water nanofluids, and concluded that nanofluids with a higher volume fraction and lower particle size are in favor of higher efficiency in a heat sink in certain conditions. Fani [38] deal with the size effect of spherical nanoparticles on thermal performance and pressure drop of a CuO–water nanofluid in a trapezoidal MCHS. The results signified that the heat transfer decreases with the increase of particle size and decreases more obviously when the volume fraction is larger.

A MCHS with periodically arranged V-ribs using a nanofluid is expected to be a good choice for micro solar cells because V-ribs can produce chaotic convection in a microchannel [39]. The other mechanisms (e.g., interruption of the boundary layer and increasing the heat transfer area) were investigated in previous work [21]. The objective of the present work is to design a promising heat sink for micro solar cells by inserting V-ribs on channel walls to enhance convective heat transfer. For this purpose, the influence regularity of V-ribs on the thermal performance of the MCHS will be explored, and then the preferential values of V-rib parameters for thermal performance by numerical simulations will be determined.

## 2. Geometrical Model 

On the basis of chaotic mixing in a microchannel proposed by Stroock [39], a MCHS with two sets of V-ribs only located on the bottom wall (heated wall, the heat flux *q* = 100 W/cm^2^), i.e., single-sided rib channel (SRC), was designed (as shown in Figure 1a). Another MCHS (Figure 1b) with V-ribs on both the top and bottom walls is an example of a double-sided rib channel (DRC). Note that the heat flux is from the micro solar cell under the MCHS (Figure 1c). Every V-rib is composed of two slant sides with different lengths, a and b. The angle between the two slant sides, named the flare angle, is α (Figure 2). In order to induce remarkable chaotic convection, every V-rib set is composed of six V-ribs, and their side lengths a and b will exchange in the succedent set of V-ribs. Every six V-ribs is defined as a half cycle. Furthermore, the height and width of the V-ribs are hr and t, respectively. The rib width is fixed at 30 μm in the present work. The pitch of the V-ribs is set to be d. The overall dimension of the MCHS for the micro solar cell is 5000 μm × 5000 μm. Each heat sink model was composed of eleven microchannels. For decreasing the computation, only one microchannel in the middle was considered in the numerical simulation. The length L, width W and height H are 5245 µm, 300 µm, 300 μm, respectively. The thickness of the bottom wall and side wall of the microchannel are fixed at 150 µm and 100 μm, respectively. The ranges of the geometrical parameters of the V-rib are listed in Table 1. Note that the rib height in the SRC was double that in the DRC. In addition, the height of the cover plate, silicon substrate and printed circuit board (PCB) plate are 150 μm, 450 μm and 150 μm, respectively. PCB is employed to connect the current from micro solar cell.

## 3. Numerical Analysis

### 3.1. Numerical Model

Considering the conjugated heat transfer of solid–fluid, heat transfer in fluids and solids should be involved. In allusion to the silicon-substrate MCHS using nanofluid as the working medium, several assumptions were made to simplify the numerical calculation, e.g., no viscous dissipation [9], incompressible and laminar flow, and the neglect of gravitation and radiation heat transfer. The governing equations for fluids are the same as the two-component four-equation model for nanofluids derived by Buongiorno [34]:

(1) Conservation of mass:(1)$$\nabla \cdot \vec{V}=0$$
where $\vec{V}$ is the velocity of the nanofluid.

(2) Conservation of momentum:(2)$$\rho_{nf}[\frac{\partial \vec{V}}{\partial t}+(\vec{V}\cdot \nabla )\vec{V}]=-\nabla p+\mu_{nf}\nabla^{2}\vec{V}$$
where $\rho_{nf}$ is the density of the nanofluid, p is pressure, and $\mu_{nf}$ is the dynamic viscosity of the nanofluid. 

(3) Conservation of energy:(3)$$\rho_{nf}c_{nf}\left[\frac{\partial T}{\partial t}+\vec{V}\cdot \nabla T\right]=\nabla \cdot k_{nf}\nabla T+\rho_{p}c_{p}[D_{B}\nabla_{\phi }\cdot \nabla T+D_{T}\frac{\nabla T\cdot \nabla T}{T}]$$
where $c_{nf}$ is specific heat and $k_{nf}$ is the thermal conductivity of the nanofluid. T is temperature. Besides, $D_{B}=\frac{K_{B}T}{3\pi \mu_{f}d_{p}}$ is the Brownian diffusion coefficient related to the nanoparticle diameter $d_{p}$, where $K_{B}=1.381\;\times \;10^{-23}$. $D_{T}=\;0.26\frac{k_{f}}{2k_{f}+k_{p}}\frac{\mu_{f}}{\rho_{f}}\phi$ represents the thermophoretic diffusion coefficient related to the volume fraction of nanoparticle φ, and $k_{f}$, $\rho_{f}$, $\mu_{f}$ are thermal conductivity, density, and dynamic viscosity of the base liquid, respectively.$k_{p}$ is the thermal conductivity of the nanoparticle. 

(4) Conservation of species:(4)$$\frac{\partial \varphi }{\partial t}+\vec{V}\cdot \nabla \varphi =\nabla \cdot [D_{B}\nabla_{\varphi }+D_{T}\frac{\nabla T}{T}]$$

For solids, only thermal conduction is taken into account. The energy conservation equation can be written as [9]:(5)$$k_{s}\nabla^{2}T=0$$
where $k_{s}$ is the thermal conductivity of the silicon substrate.

### 3.2. Parameter Settings

The thermal physical parameters for the Al_2_O_3_–water nanofluid are determined according to that of the base liquid and nanoparticles (Table 2). The dynamic viscosity of the base fluid related to the temperature is read as [40]:(6)$$\mu_{f}=(2.414\times 10^{-5})\times 10^{\frac{247.8}{\left(T-140\right)}}$$

Based on the Brinkman model, the dynamic viscosity of the nanofluid is [40]:(7)$$\mu_{nf}=(1+2.5\varphi )\mu_{f}$$

The density and heat specific of the nanofluid can be written as [41]:(8)$$\rho_{nf}=\varphi \rho_{p}+(1-\varphi )\rho_{f}$$
(9)$${\left(c\rho \right)}_{nf}=\varphi {\left(c\rho \right)}_{p}+(1-\varphi ){\left(c\rho \right)}_{f}$$

The thermal conductivity model of Chon [42], which is in line with the experimental results of Al_2_O_3_–water nanofluids, can be read as:(10)KnfKf=1+64.7φ0.75(dfdp)0.37(kpkf)0.75Pr Rep1.23
where the diameter of the water molecule $d_{f}=$ 0.28 nm, Prandtl number $Pr=c_{f}\mu_{f}/k_{f}$, particle Reynolds number $Re_{p}=\rho_{f}u_{B}d_{p}/\mu_{f}$, Brownian velocity $u_{B}=k_{B}T/3\pi \mu_{f}d_{p}\lambda_{f}$, and the mean free path of water molecule $\lambda_{f}$ is set to be 0.17 nm [43].

### 3.3. Validity of the Numerical Model

The purpose of the present work is to designe a MCHS with good thermal performance for micro solar cells. Hence, numerical simulations were carried out to study the flow and heat transfer in microchannels with V-ribs for a reasonable arrangement of V-ribs and preferential geometric parameters of V-ribs.

The process of numerical simulation is: (1) geometric modeling, (2) meshing, (3) setting up the boundary condition, (4) compiling the user defined function according to Equations (7)–(10), (5) solving by Fluent 12.1, (6) data outputting, and (7) post-processing by Tecplot 10.0. In the simulation, no velocity–slip and no temperature–jump boundary conditions are enforced, uniform “inlet-velocity” $v_{in}$ in the direction of +x is assumed, and “outflow” occurs at the outlet. The initial temperature for both the fluid and solid is 300 K. A constant heat flux from the surface of the solar cell is assumed according to [44]. Hence, Riemann boundary condition $-k_{s}\frac{\partial t}{\partial y}=q$ should be exerted on the bottom wall, where q = 100 W/cm^2^ on the bottom wall, and “adiabatic”, namely, *q* = 0 on all other walls, at inlets and at outlets. All contact surfaces between different materials are set to be “interface”. In consideration of the interplay among physical quantities such as volume fraction, temperature, the viscosity of the nanofluid, the motion of nanoparticles in the Al_2_O_3_–water nanofluid, the related characteristic parameters (such as viscosity, density, specific heat capacity, and thermal conductivity of nanofluid), Brownian diffusion and the thermophoretic diffusion coefficient will be affected. Subsequently, the user defined function programmed from Equations (7)–(15) should be invoked in every iteration.

Verification of grid independence was conducted. The geometric parameters of the V-rib are set to be t/W=0.1, d/W=1, α=75°, hr/H=0.5, b/a=0.5, respectively. The discrepancy of the calculated Nusselt number for grid numbers 62,156, 123,850, 261,253 and 415,622 were 3.13%, 1.58%, 0.11%, respectively, when the Reynolds number was 500. The calculation accuracy is high enough when grid numbers are larger than 261,253.

It is suggested that the validity check of the numerical model needs to be carried out before being employed in practical engineering. Numerical simulations for the friction coefficient and Nusselt number in such a MCHS were conducted. Good agreements shown in Figure 3 accomplish the validity check of the numerical model by comparing the numerical results of the present model with the published results in. [45].

## 4. Results and Discussion

It is known from most existing MCHSs that the width–height ratio W/H of the microchannel is generally smaller for larger cooling areas because more identically-sized side walls can be arranged in the MCHS. The presented MCHS with V-ribs has a greater W/H=1 for a smaller pressure drop. Meanwhile, the cooling area and pressure drop can both be increased by the inserted V-ribs. However, interruption of the thermal boundary and production of chaotic convection by the inserted V-ribs can elevate the heat transfer performance of the MCHS. As a result, better thermal performance of a MCHS with V-ribs can be expected.

### 4.1. Definition of Relevant Parameters

In order to assess the heat transfer performance of a MCHS, it is necessary to define a suitable non-dimensional group. The Nusselt number (Nu) represents the heat transfer performance caused by convection and is defined as [35]:(11)$$Nu=\frac{hD}{k_{nf}}$$
where D is the hydraulic diameter, and the heat transfer coefficient h can be written as:(12)$$h=\frac{qA_{q}}{A_{c}\left(T_{c}-T_{f}\right)}$$
where $A_{q}$, $A_{c}$ are the heated area and the conjugated area (i.e., the solid–fluid interface area on the bottom wall), respectively. $T_{c}=\frac{\int TdA}{\int dA}$ denotes the average temperature in the conjugated area, $T_{f}=\frac{\int T\rho_{f}dV}{\int \rho_{f}dV}$ represents the average temperature of fluid.

The friction coefficient (denoting resistance force) can be defined as:(13)$$f=\frac{2\Delta pD}{L\rho_{f}v_{in}^{2}}$$
where Δp is the pressure drop.

The pump power of the MCHS for steady flow can be defined as:(14)$$P_{p}=\Delta pv_{in}A_{in}$$
where inlet area $A_{in}=\frac{\varphi^{2}\pi }{4}$, ϕ is the inlet diameter (set to be 330 μm in the present work). The average Nusselt number with identical pump power is suggested to be a suitable index to evaluate the thermal performance of MCHS [9] and can be employed in the present work. In fact, it can be found after careful analysis that the Nusselt number with identical pump power is equivalent to the performance evaluation criteria (PEC) proposed in [46].

### 4.2. Effect of Volume Fraction of Nanoparticles

It is known that the nanofluid in a MCHS can enhance heat transfer due to the elevation of thermal conductivity as a result of the added nanoparticles. In fact, the mechanisms of heat transfer enhancement by nanofluids is complicated and comprehensive. Buongiorno [34] suggested the most important mechanisms are Brownian motion and thermophoresis of nanoparticles. Hence, numerical simulations were conducted in both SRC and DRC to calculate the Nusselt number and friction coefficient when the Reynolds number ranges from 200 to 1000. The numerical results shown in Figure 4 indicate that the Nusselt number and friction coefficient increase with Reynolds number, but their tendencies are different. The Nusselt number increases sharply at lower Reynolds numbers and tends to flat at higher Reynolds numbers. Inversely, the friction coefficient does not increase at lower Reynolds numbers and increase violently at higher Reynolds numbers. Therefore, it can be concluded that a moderate Reynolds number should be most suitable for thermal performance. In addition, Figure 4 shows that a greater volume fraction of nanoparticle gives rise to a greater Nusselt number and friction coefficient. The reasons for this are greater thermal conductivity and greater viscosity.

### 4.3. Overall Flow Patterns in a Microchannel

The flow patterns in SRC and DRC can be obtained from the numerical simulations. The parameters are set to be: *Re* = 600, d/W=1, α=75°, hr/H=0.5, and b/a=0.4. It can be seen from Figure 5a,b that not all streamlines in DRC are helical; instead, only the streamlines in the bottom half are helical because only V-ribs on bottom wall are inserted. It is known from mixing theory [47] that helical streamlines can stretch and fold the fluids to enhance mixing between a hot fluid and a cool fluid. Hence, it can be expected that the thermal performance of DRC is superior to that of SRC.

### 4.4. Overall Flow Patterns in Microchannel

The flow patterns in SRC and DRC can be obtained from the numerical simulations. The parameters are set to be: *Re* = 600, d/W=1, α=75°, hr/H=0.5, b/a=0.4. It can be seen from Figure 5a,b that, all streamlines in DRC are helical, instead, only the streamlines in bottom half are helical because only V-ribs on bottom wall are inserted. It is known from mixing theory [47], the helical streamlines can stretch and fold the fluids to enhance mixing between hot fluid and cool fluid. Hence, it can be expected, the thermal performance of DRC is superior to that of SRC.

To gain insight into the heat transfer performance of a MCHS, the temperature distributions in the microchannel should be calculated (Figure 6a–f). Comparison of Figure 6a and Figure 6b indicates that the right vortex induced by the long side of the V-rib a can visibly take some heat away from the bottom wall to a higher position. However, the left one does not affect the heat transfer as visibly. This is because the right vortex is larger than the left one. Comparing Figure 6e and Figure 6d shows that the heat transfer enhancement improves very sharply by the seventh V-rib because the asymmetric arrangement of V-ribs (sixth and seventh V-rib) shifts the center of the flow patterns. The underlying mechanism is the shift of the center of the flow pattern to produce blinking flow and consequently induce chaotic convection [47]. Comparing Figure 6d and Figure 6f shows that the heat transfer performance can be further improved on by succedent V-ribs (eighth to twelfth V-rib).

To insight into the heat transfer performance of such a MCHS, the temperature distributions in microchannel should be calculated (Figure 7a–f). Comparison of Figure 7a and Figure 7b indicates that, the right vortex induced by the long side of V-rib a can visibly take some heats away from the bottom wall to a higher position, the left one, instead, doesn’t affect the heat transfer so visibly. This is because the right vortex is larger than the left one. Comparing of Figure 7e and Figure 7d shows that, the heat transfer enhancement improves very sharply by 7th V-rib. Because the antisymmetric arrangement of V-ribs (6th and 7th V-rib) shifts the center of flow patterns. The underneath mechanism is the shift of center of flow pattern produce blinking flow and consequently induce chaotic convection [47]. Comparison of Figure 7d and Figure 7f indicates that, the heat transfer performance can be further improved on by succedent V-ribs (8th–12th V-rib).

In order to clarify the mechanism of heat transfer enhancement by the V-ribs, the temperature fields and flow fields in a plane perpendicular to the main flow are acquired from the numerical simulations. The parameter settings are the same as aforementioned. Figure 8 shows the temperature profiles along with velocity vectors in the normal plane compared to the main stream within the region of the fifth V-rib. It can be seen that the vortexes decline in front of the fifth V-rib. The left one in particular almost disappears (Figure 8a). However, the larger vortex from the right half of the channel is driven to the left half and swallows up the recession vortex in the left half (Figure 8b). There is no apparent vortex in the right half of the channel because of the suppression of both sides of the V-rib (Figure 8c,d). The fluid temperature in the region between both sides of the V-rib remains high (Figure 8c,d). The reason for this is that the heat in this region cannot be taken away so easily due to no vortexes, and therefore more heat can be transferred from the rib to the fluid. In addition, the cool fluid from the center of the upper half of the channel can be dragged to the bottom by the vortexes, and the heat transfer efficiency increases consequently (Figure 8b,c). What has changed is that a smaller vortex forms near the right side wall owing to the disappearance of the short side of V-rib b(Figure 8e), and the larger vortex is found in the right region between the two sides of the V-rib (high temperature region). This is beneficial to elevate heat transfer efficiency. It can be seen from Figure 8f that the cool region shrinks compared to Figure 8e. This means that the heat transfer efficiency is elevated by the arranged V-ribs. 

### 4.5. Planar Flow Patterns in the DRC

Similarly, the planar flow patterns in the DRC should be acquired from numerical simulations to uncover the underlying mechanism of heat transfer enhancement in MCHS with DRC. For the same spiral direction of streamlines in both the top and bottom half of the channel, V-ribs and Λ-ribs (i.e., inverted V-ribs) are inserted on the top and bottom wall, respectively. Two vortex-pairs can be seen in the plane normal to the main stream at the rear of the first V-rib; that is, there are four pattern centers in right-top, right-bottom, left-top and left-bottom of the channel (Figure 9a). But, the left vortex-pair is smaller than left one, because the length of the left side of the V-rib, a, is larger than that of the right side, b. On this occasion, the hot fluid near bottom wall can be brought to the channel center, and the cool fluid near the top wall is instead driven down to the channel center as well. The right vortex-pair emerges gradually into a larger vortex from the second V-rib to the fifth V-rib (Figure 9b–e). Note that a great change in flow patterns can be seen in Figure 9d,f. The positions of the larger vortex-pair and smaller vortex-pair shift from right to left. Figure 9e shows the transitional flow pattern without an obvious vortex-pair. The planar flow patterns in the rear of the eighth to twelfth V-rib are altered in the same manner as described above. The larger vortex-pair emerges into one vortex gradually, whereas the smaller vortex splits into a vortex-pair gradually (Figure 9h–k). This shift of the centers of the flow patterns can induce chaotic convection, and consequentially greatly improve the heat transfer efficiency in MCHS with DRC.

To investigate the heat transfer efficiency in a MCHS with DRC, the temperature profiles should be calculated (Figure 10a–f). The comparison of Figure 10a and Figure 10b implies that the right vortex-pair can visibly take some heat away from the bottom and side walls to a central position because of the larger vortex-pair in right half of channel. The left one does not affect the heat transfer as visibly. The heat transfer efficiency in the rear of the sixth V-rib (Figure 10c) is greatly improved compared to that in rear of the first V-rib (Figure 10b). In addition, the heat transfer enhancement improves significantly by the seventh V-rib (Figure 10d) because the second set of V-ribs is reversely arranged compared to the first one. The underlying mechanism is the shift in the center of the flow pattern which can produce blinking flow and consequently induce chaotic convection [47]. A comparison of Figure 7d and Figure 7f indicates that the heat transfer performance can be further improved by succedent V-ribs (eighth to twelfth V-ribs).

Why can the inserted V-ribs improve heat transfer performance in a MCHS with DRC? Figure 11 shows the temperature profiles along with velocity vectors during the fifth V-rib. It can be seen that heat transfer enhancement by V-ribs is obvious when comparing Figure 11a and Figure 11f. This is attributed to a shift in the flow patterns and the resulting chaotic convection under the action of the ribs. In detail, the left vortex-pair fuse into a larger vortex in front of the fifth V-rib, and the right vortex-pair is in the process of fusion (Figure 11a). The fused vortex in the left half of the channel can be stretched by the V-ribs on the top and bottom walls and extends to right-top. The right vortex-pair is pressed down to right-bottom by the stretched vortex (Figure 12b). The vortex extending to right-top will be split by the V-rib on the top wall and will gradually form a new vortex-pair (Figure 12c–e). Finally, a newly-generated vortex at the left-bottom corner (Figure 12e) will be swallowed up by the larger vortex which occupies the left half of the channel. Consequently, two vortex-pairs form again (Figure 9d and Figure 12f). 

### 4.6. Comparison of Temperature at Various Heights

Temperature profiles at various heights (*z*) for both SRC and DRC are shown in Figure 12. It can be seen that there is no distinct difference in the vicinity of the bottom wall (Figure 12a), whereas the temperature distribution uniformity (Figure 12b,c,e) of SRC is better than that of DRC in the region of the channel center (i.e., *z =* 70 µm, 140 µm, 210 μm). This is because larger vortexes (Figure 6) induced by higher ribs in the SRC can produce more violent chaotic convection than those in DRC. Analogous situations exist in the region 75 μm<z<225 μm owing to no ribs. On the contrary, the uniformity of temperature (Figure 12e) for SRC is obviously worse than that for DRC in the top half of the channel (*z* = 270 μm) because a number of V-ribs which can induce vortexes are inserted there on the top wall of the microchannel. In summary, it can be concluded that the heat transfer performance of DRC is better than that of SRC when comparing Figure 7, Figure 10 and Figure 12. 

### 4.7. Parameterized Study of V-ribs

It is known that the geometric parameters will affect heat transfer. The geometric parameters of V-ribs that affect the heat transfer performance of MCHS involve flare angle, rib height, rib pitch and the lengths of the slant sides of V-ribs. Some non-dimensional parameters are defined to clarify the heat transfer rule in a MCHS. The flare angle of the V-rib is α. The rib height is represented by the ratio of rib height to channel height hr/H. The rib pitch can be expressed as the ratio of rib pitch to channel width d/W. The ratio b/a represents the asymmetry of the V-rib. The positive influence of V-ribs on heat transfer in MCHS is known, and the pressure drop increases accordingly. PEC and the average Nusselt number with identical pump power are two proposed indexes for assessing the heat transfer performance of MCHS [9,46]. Numerical simulations for various flare angles of V-ribs at various Reynolds numbers (200–1000) are carried out, and the average Nusselt numbers ($Nu/Nu_{0}$) and friction coefficients ($f/f_{0}$) are calculated, where $Nu_{0},f_{0}$ are the Nusselt number and friction coefficient for the MCHS without V-ribs. Figure 13a indicates that $Nu/Nu_{0}$ increases with the Reynolds number when the Reynolds number is less than a critical value, and drop when the Reynolds number is larger than the critical value for all flare angles. The critical value of Reynolds number is about 600 and 400 for DRC and SRC, respectively. It is worth noting that the $Nu/Nu_{0}$ in DRC are larger than in SRC because more intense convection can be induced by the combined action of V-ribs located on both the top and bottom wall. Figure 13b shows that the friction coefficients always increase with Reynolds number. Note that $f/f_{0}$ in DRC is larger than that in SRC when the flare angle is larger than 75°, and the reverse is true when the flare angle is less than 75°. After comprehensive consideration of $f/f_{0}$ and $Nu/Nu_{0}$, it is known that Nu increases steeply with the pump power at lower $P_{p}$, but it increases gradually at larger $P_{p}$ (Figure 13c). It is considered that a larger Nusselt number at an identical pump power means better heat transfer performance. Hence, the preferential value of the flare angle can be seen in Figure 12c: 75° for DRC and 60° for SRC.

Analogously, the effect of rib height on heat transfer is investigated as well (see Figure 14). The preferential value of rib height is hr/H=0.3 for both DRC and SRC. The effect of rib height on heat transfer can be interpreted as follows: a smaller rib height can produce chaotic convection only in the vicinity of inserted V-ribs; contrarily, a larger rib height results in a greater pressure drop. So, a medium rib height is beneficial to enhance heat transfer in MCHS. Note that the heat transfer performance of MCHS with very low rib height hr/H=0.1 can be improved by about 124%–202% compared to that without ribs (hr/H=0). This is because the thermal boundary is interrupted by the inserted ribs.

Another important factor that influences the heat transfer performance of MCHS is rib pitch d/W. It is conceivable that the rib pitch should match with the flare angle and the velocity of the main flow. Figure 15 indicates that d/W=1 can be regarded as the preferential value for both SRC and DRC.

Finally, the asymmetric V-ribs can result in two vortexes in SRC and two vortex-pairs in DRC. Blinking flow can easily be induced by such asymmetric V-ribs because of the shift of flow patterns. Figure 16 shows that b/a=0.75 is more preferential for both SRC and DRC. It can be seen in Figure 6 and Figure 9 that asymmetric V-ribs can intensify chaotic convection. In addition, smaller vortexes or vortex-pairs are easily declined when b/a=0.25, 0.5, as the fluid circulation is weakened which consequently degrades any heat transfer enhancement.

## 5. Conclusions

On the basis of chaotic mixing theory, a new microchannel heat sink with V-ribs is designed and numerically simulated. The flow patterns and temperature distributions are analyzed to uncover the underlying mechanism of heat transfer enhancement in such a MCHS. Moreover, preferential values of geometric parameters are determined.

MCHS with V-ribs can elevate the heat transfer performance because of the chaotic convection in the microchannel caused by the inserted V-ribs. The interruption of the thermal boundary and an increase in the cooling area are two other important mechanisms. The heat transfer performance of MCHS with ribs can be improved by about 124%–202% compared to a MCHS without ribs.

The thermal performance in DRC is better than in SRC because more intense chaotic convection can result when DRC are arranged in a MCHS.

Preferential values of geometric parameters were determined numerically. For SRC, d/W=1, α=60°, hr/H=0.3, b/a=0.75. For DRC, d/W=1, α=75°, hr/H=0.3, b/a=0.75.

## Figures and Tables

**Figure 1 micromachines-10-00620-f001:**
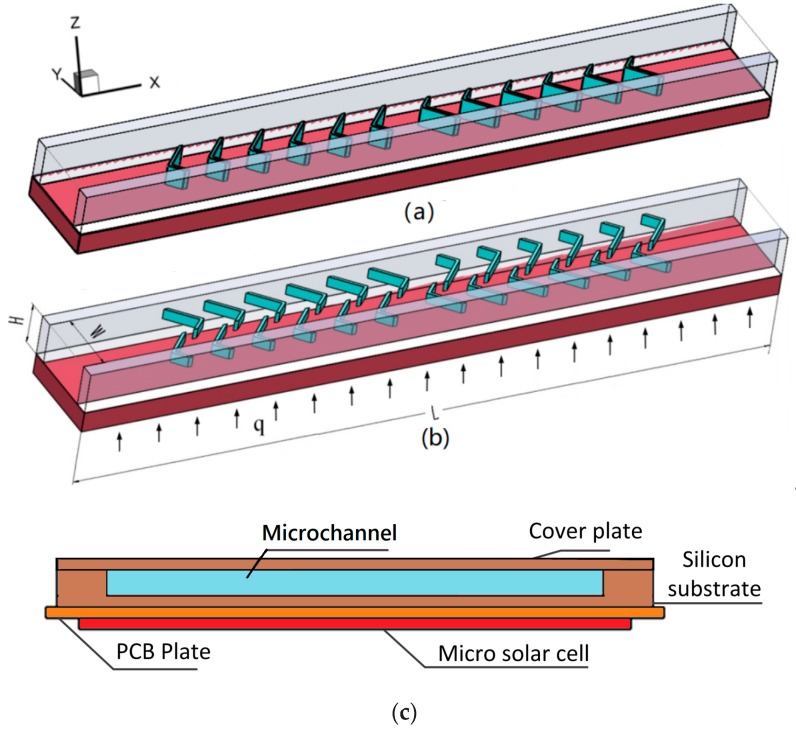
Geometrical structure of microchannel with V-ribs. (**a**) Single-sided rib channel; (**b**) double-sided rib channel; (**c**) microchannel heat sink (MCHS) and micro solar cell.

**Figure 2 micromachines-10-00620-f002:**
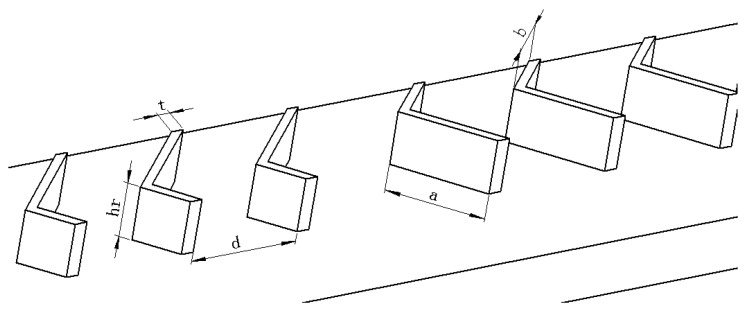
Geometrical parameters of V-ribs. Slant sides length a and b, Rib height hr, rib width d, rib thick t.

**Figure 3 micromachines-10-00620-f003:**
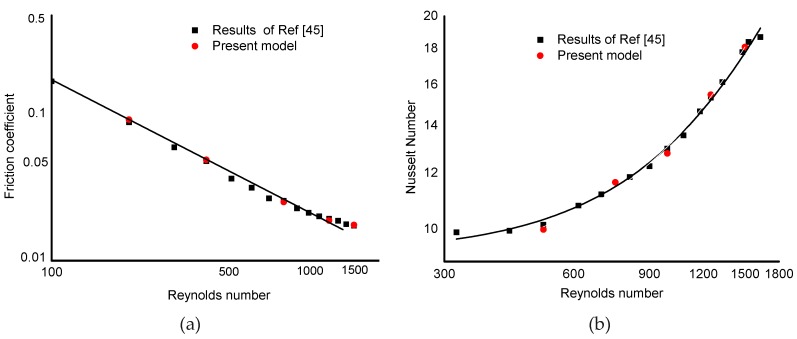
Validity check of present model. (**a**) Friction coefficient vs. Reynolds number, (**b**) Nusselt number vs. Reynolds number.

**Figure 4 micromachines-10-00620-f004:**
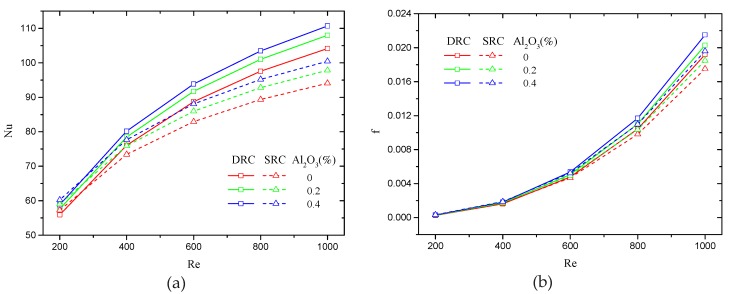
Effect of volume fraction and Reynolds number on Nusselt number. (**a**) Friction coefficient (**b**) in a single-sided rib channel (SRC) and a double-sided rib channel (DRC). The geometric parameters are d/W=1, α=75°, hr/H=0.5, and b/a=0.5.

**Figure 5 micromachines-10-00620-f005:**
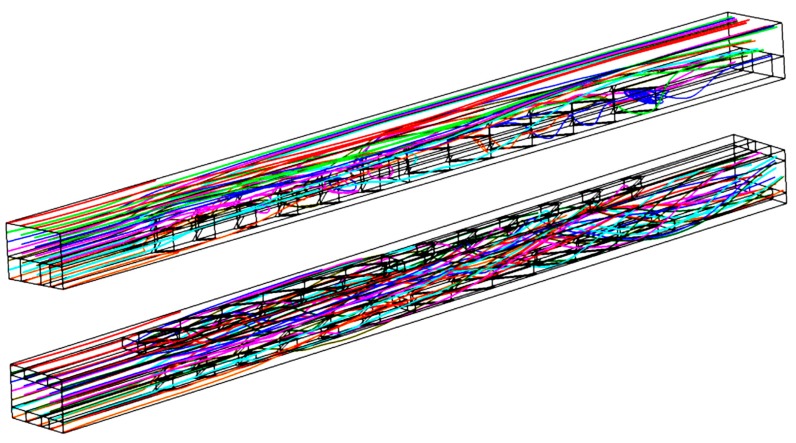
Overall flow patterns in a microchannel. *Re* = 600, d/W=1, α=75°, hr/H=0.4, b/a=0.5 for SRC (top) and DRC (bottom).

**Figure 6 micromachines-10-00620-f006:**
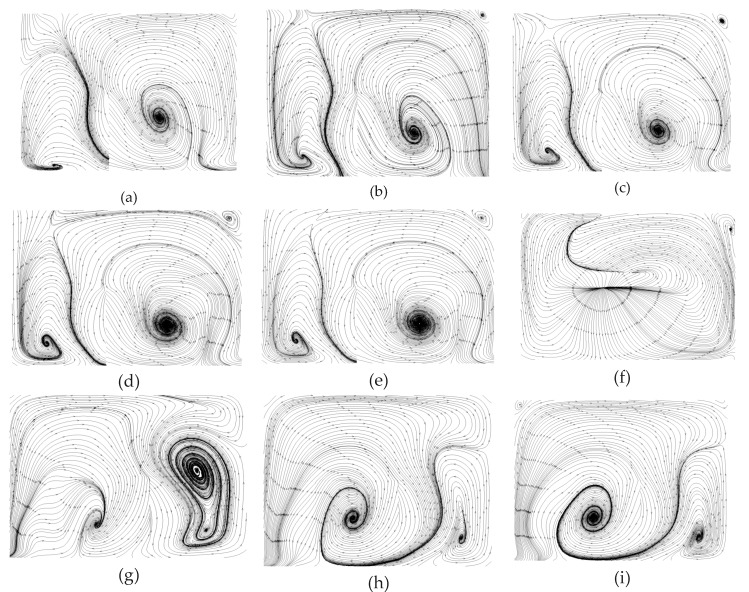
The variations of flow patterns consisting of streamlines at the plane normal to the main stream due to the perturbation of V-ribs. The geometric parameters are *Re* = 1000, d/W=1, α=75°, hr/H=0.5, b/a=0.5. (**a**) Rear of first V-rib, *x* = 300 μm, (**b**) rear of second V-rib, *x* = 600 μm, (**c**) rear of third V-rib, *x* = 900 μm, (**d**) rear of fourth V-rib, *x* = 1200 μm (**e**) rear of fifth V-rib, *x* = 1500 μm, (**f**) rear of sixth V-rib, *x* = 1800 μm, (**g**) rear of seventh V-rib, *x* = 2200 μm, (**h**) rear of eighth V-rib, *x* = 2500 μm, (**i**) rear of ninth V-rib, *x* = 2800 μm, (**j**) rear of tenth V-rib, *x* = 3100 μm, (**k**) rear of eleventh V-rib, *x* = 3400 μm, (**l**) rear of twelfth V-rib, *x* = 3700 μm.

**Figure 7 micromachines-10-00620-f007:**
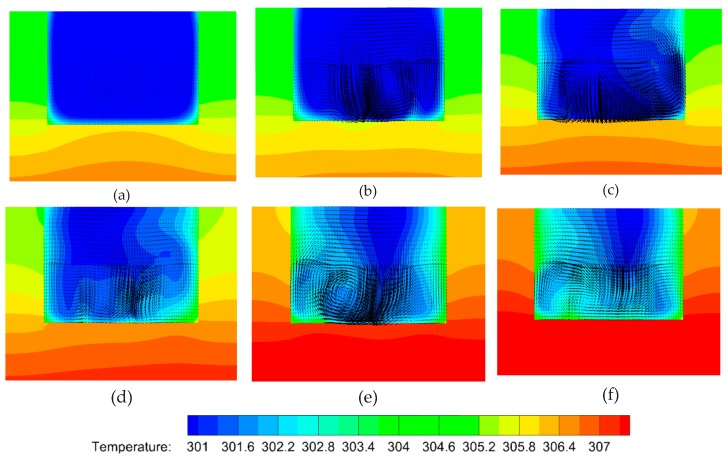
The temperature profiles and velocity vectors at various planes normal to the main stream due to the V-ribs. The geometric parameters are *Re* = 1000, d/W=1, α=75°, hr/H=0.5, and b/a=0.5. (**a**) Front of first V-rib, *x* = 200 μm, (**b**) rear of first V-rib, *x* = 300 μm, (**c**) rear of sixth V-rib, *x* = 1800 μm, (**d**) rear of seventh V-rib, *x* = 2200 μm, (**e**) rear of eleventh V-rib, *x* = 3400 μm, (**f**) rear of twelfth V-rib, *x* = 3700 μm.

**Figure 8 micromachines-10-00620-f008:**
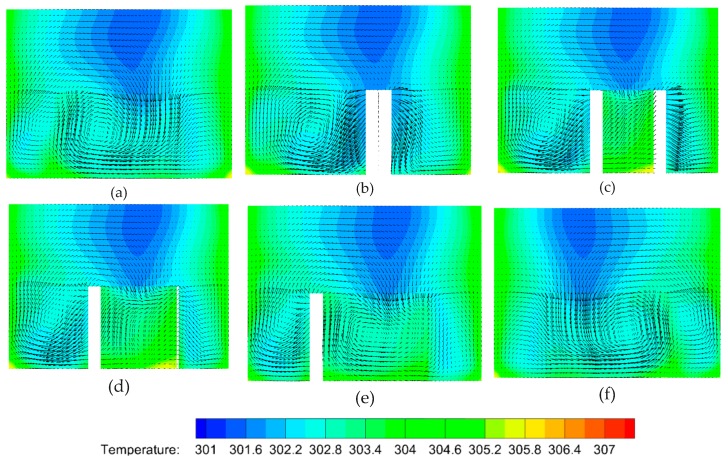
Temperature profiles with velocity vectors in various planes perpendicular to the main flow in the region of the fifth V-rib. The parameters are *Re* = 1000, d/W=1, α=75°, hr/H=0.5, b/a=0.5. (**a**) *x* = 3050 μm, (**b**) *x* = 3140 μm, (**c**) *x* = 3200 μm, (**d**) *x* = 3230 μm, (**e**) *x* = 3290 μm, (**f**) *x* = 3340 μm.

**Figure 9 micromachines-10-00620-f009:**
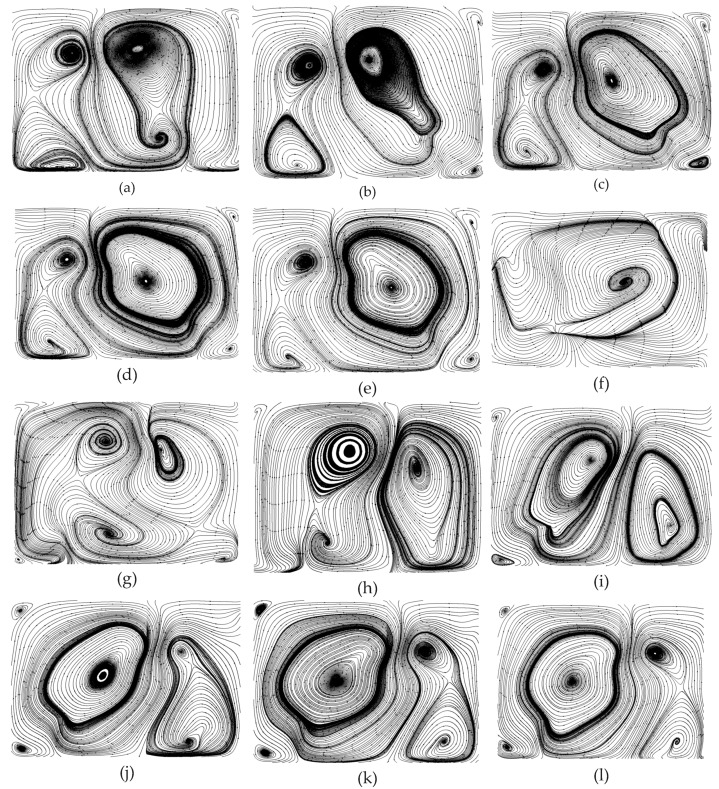
The variations of flow patterns consisting of streamlines at various planes normal to the main stream because of the perturbation of V-ribs. The geometric parameters are *Re* = 1000, d/W=1, α=75°, hr/H=0.5, b/a=0.5. (**a**) Rear of first V-rib, *x* = 250 μm, (**b**) rear of second V-rib, *x* = 550 μm, (**c**) rear of third V-rib, *x* = 850 μm, (**d**) rear of fourth V-rib, *x* = 1150 μm, (**e**) rear of fifth V-rib, *x* = 1450 μm, (**f**) rear of sixth V-rib, *x* = 1800 μm, (**g**) rear of seventh V-rib, *x* = 2150 μm, (**h**) rear of eigth V-rib, *x* = 2450 μm, (**i**) rear of ninth V-rib, *x* = 2750 μm, (**j**) rear of tenth V-rib, *x* = 3050 μm, (**k**) rear of eleventh V-rib, *x* = 3350 μm, (**l**) rear of twelfth V-rib, *x* = 3650 μm.

**Figure 10 micromachines-10-00620-f010:**
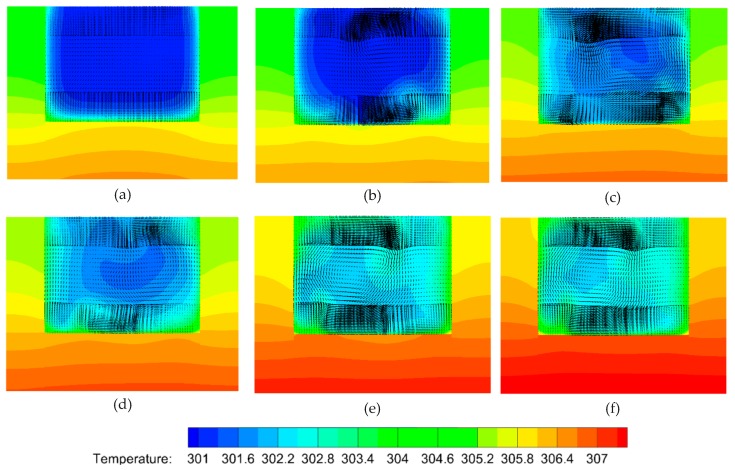
The temperature profiles with velocity vectors in various planes normal to the main stream due to the V-ribs. The geometric parameters are *Re* = 1000, d/W=1, α=75°, hr/H=0.5, b/a=0.5. (**a**) Front of the first V-rib, *x* = 200 μm, (**b**) rear of the first V-rib, *x* = 300 μm, (**c**) rear of the sixth V-rib, *x* = 1800 μm, (**d**) rear of the seventh V-rib, *x* = 2200 μm, (**e**) rear of the eleventh V-rib, *x* = 3400 μm, (**f**) rear of the twelfth V-rib, *x* = 3700 μm.

**Figure 11 micromachines-10-00620-f011:**
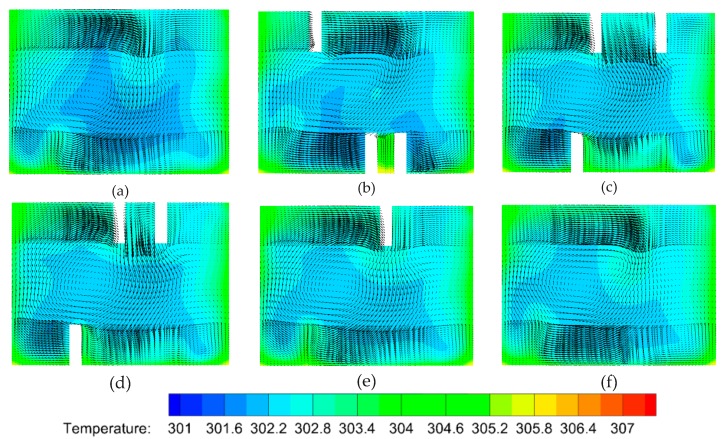
The temperature profiles along with velocity vectors in various planes perpendicular to the main flow in the region of the fifth V-rib. The parameters are *Re* = 1000, d/W=1, α=75°, hr/H=0.5, b/a=0.5. (**a**) *x* = 3110 μm, (**b**) *x* = 3160 μm, (**c**) *x* = 3250 μm, (**d**) *x* = 3280 μm, (**e**) *x* = 3300 μm, (**f**) *x* = 3330 μm.

**Figure 12 micromachines-10-00620-f012:**
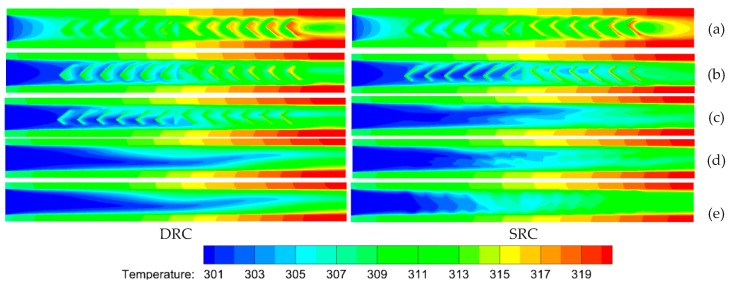
Comparison of temperature profiles at various heights (z) between SRC and DRC. *Re* = 1000, d/W=1, α=75°, hr/H=0.5, b/a=0.5. (**a**) *z* = 10 μm, (**b**) *z* = 70 μm, (**c**) *z* = 140 μm, (**d**) *z* = 210 μm, (**e**) *z* = 270 μm.

**Figure 13 micromachines-10-00620-f013:**
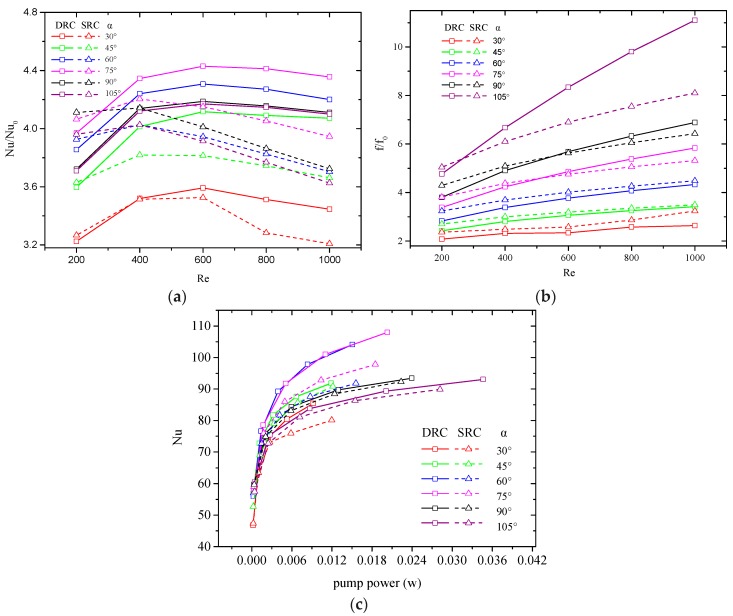
Effect of flare angle on the Nusselt number and friction coefficient. d/W=1, hr/H=0.5, b/a=0.5. (**a**) $Nu/Nu_{0}$ vs. Re, (**b**) f vs. Re, (**c**) Nu vs. $P_{p}$.

**Figure 14 micromachines-10-00620-f014:**
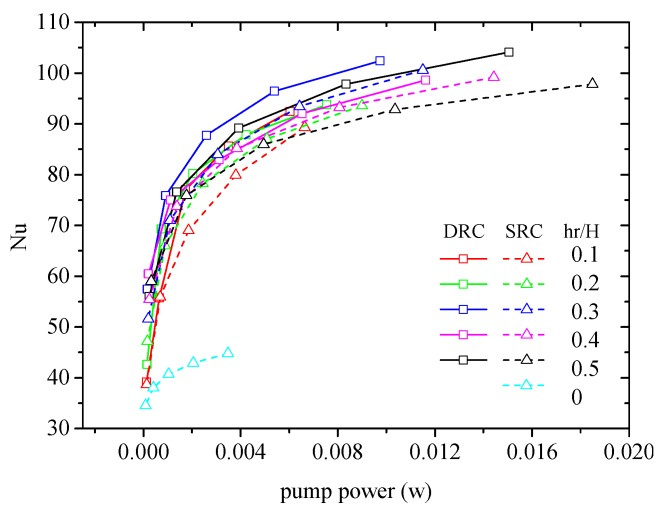
Effect of rib height on the heat transfer enhancement in MCHS. d/W=1, α=75°, b/a=0.5.

**Figure 15 micromachines-10-00620-f015:**
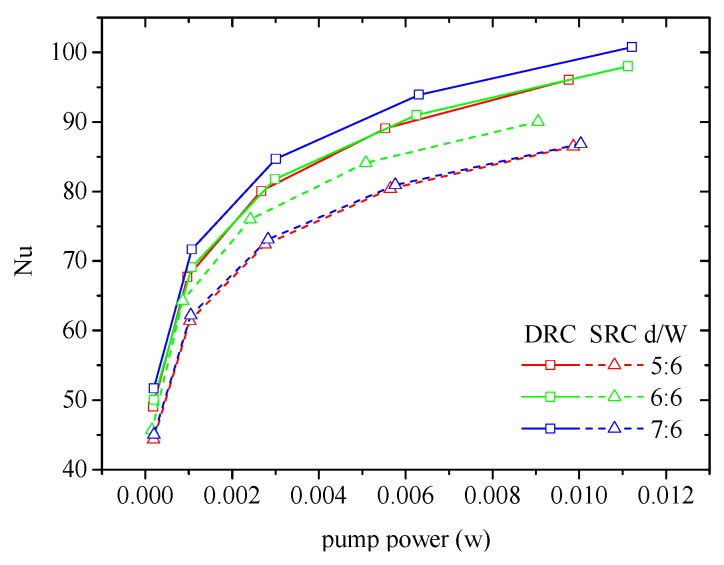
Effect of rib pitch on the heat transfer in MCHS. α=75°, hr/H=0.3, b/a=0.5, *Re* = 600 for DRC and *Re* = 400 for SRC.

**Figure 16 micromachines-10-00620-f016:**
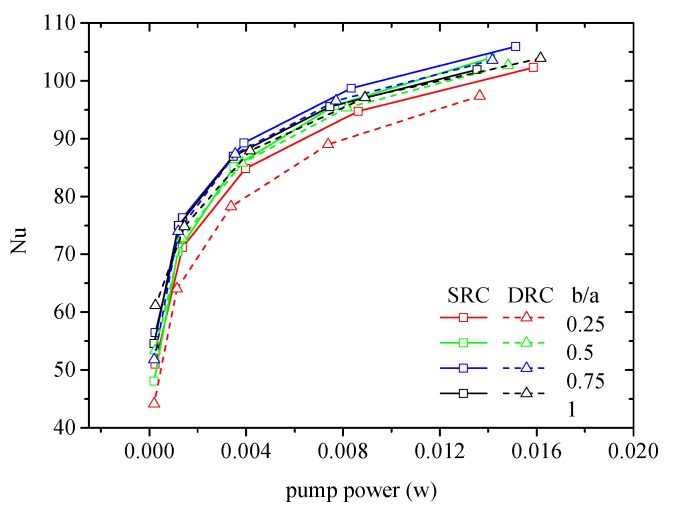
Effect of asymmetric ribs on the heat transfer in MCHS. α=75°, d/W=1, hr/H=0.3.

**Table 1 micromachines-10-00620-t001:** Range of geometric parameters of the V-rib used in simulations.

Ratio of Side *a/b*	Flare Angle *α* (°)	Rib Height *hr/H*	Rib Thickness *t/W*	Rib Pitch *d/W*
0.25–1	30–75	0.1–0.5	0.1	5/6–7/6

**Table 2 micromachines-10-00620-t002:** Thermal physical parameters for the base liquid and nanoparticles.

Parameters	$\rho \;\left({\mathrm{kg}/m}^{3}\right)$	k (W/m⋅K)	$c_{p}\;\left(J/\mathrm{kg}\cdot K\right)$
Water	998.2	0.6	4182
Al_2_O_3_	3970	42	880

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
