# Peer review of "Analysis and Optimization of a Microchannel Heat Sink with V-Ribs Using Nanofluids for Micro Solar Cells"

_micromachines, 2019, doi:10.3390/mi10090620_

Round 1
Reviewer 1 Report
This work regards the use of flow inserts in a solar system for enhancing the flow conditions. In my opinion, this work is not of high-quality and thus it cannot be published in the Journal. Below, I explain briefly my opinion:
1) The authors claim the investigation of a solar system but they do not do this. They make a flow analysis so the title and the general concept is not satisfied.
2) There is not the proper validation of the model; the proper validation needs Nusselt and friction factor validation.
3) There is not a proper evaluation of the results. The authors had to use the collector efficiency to show performance enhancement. Moreover, they had to use the PEC criterion [PEC=(Nu/Nu0)/(f/f0)^(1/3)] in order to evaluate properly the high-pressure drop increase.
4) Presenting equations in the results section is wrong, they have to be in methods.
5) The novelty of the paper is not properly given.
6) Merging many references together is not permitted.
7) Figures 16 to 12 are not so informative; these figures with similar images are needed.
8) There are two equations for viscosity, something that creates confusion.
9) The abstract and conclusions need numerical results.
Author Response
Refer to the enclosure!

Reviewer 2 Report
The paper is well written. It is suitable for this journal, however:
Where 'Fig. 4' is?
Good job!!!
Author Response
Resopnse:
The paper is well-written, it is suitable to this Journal. However,
where "Fig,4" is ?
Response: Figure 4 is cited!
Good job!
Response:Thanks
Reviewer 3 Report
To author
The manuscript topic is interesting in the heat transfer subject. However
the manuscript is not well written and formatted and here are my comments:
1. The manuscript contribution and its novelty is not clear in the introduction
and need to be identified in the last paragraph of the introduction.
2. The overall languages has to be checked by a native language person.
3. The language in the results and discussion section is very poor and needs to
be re-written in journal style.
4. The journal format has to be followed such as in the figure caption.
5. All parameters in equations shall be defined.
6. The initial and boundary conditions have to be defined.
7. The grid independence study needs to be investigated.
8. Some terms are not clear such as negraphy, index (line 364), analogues
(line 382), chaotic (line 384), blinking flow ( line 398), declined ( line
402).
9. The authors need to give references for equations for all equations.
10. Section 4.4 couldn't be reviewed by the reviewer due to non clarity, confusing and very difficult to follow. So, after incorporating all the comments this will be checked.
11. All figures need to be referred to the related paragraph.
12. The authors have to refer the figure part such as Figure 6(a) whenever the authors want to explain any figure.
13. Figure 5 is not found and needs to be added.
14. The authors need to give numbers in their discussion and not just explaining the patterns.
15. Figure 12 numbering is repeated, so all
figures captions and references need to be updated.
16. Section 2, can the authors draw a schematic of the proposed heat sink showing
the solar cell.
17. Define what is (a) and (b) in the figure 1 caption.
18. The authors shall give the list of the assumptions of the modeling and also
the considerations of taking one channel.
19. Figure 2 is not shown Figure 2(b).
20. Name of the simulation tool should be mentioned.
21. Line 124: why did you conclude it is a laminar flow?
22. Lines 159-161, not clear paragraph and needs to be re-written.
23. Lines 168-170, not clear sentence.
24. Figure 3, y-axis unit has to be between brackets.
25. Key figure need to be added to figures from Figure 6 to figure 12.
26. Figure 7, why Re=600 whereas all cases are of Re=1000.
27. Paragraphs (lines 260-280), (lines 326-339) are not clear and needs to be re-written.
28. Line 281, clouds term is not normally used as a technical in this types of
figure.
29. Heading 4.6, what is z? You can give name instead of letter. This is
applicable for the entire manuscript.
30. Lines 350-351, you need to give numbers in order to conclude which one is
efficient.
31. Line 375, explain why?
32. Line 388, what is he?
33. Line 406, the " chaotic mixing theory" is not explained any where
in manuscript, what is it?
Author Response
Refer to attachment!

Round 2
Reviewer 1 Report
This work has to be rejected for the following reasons:
1) It is not possible to study a flow phenomenon and to claim that it is for a solar application without studying the solar system. It is not a deep scientific work.
2) The validation has to be done with Nusselt number, friction factor and the solar collector efficiency.
3) The PEC parameter and the collector efficiency have to be given. Which is the solar collector efficiency enhancement?
4) Merging many references is not permitted.
5) Using symbols in the abstract is not a good and informative way for presenting your work.
Author Response
To Reviewer 1
1) It is not possible to study a flow phenomenon and to claim that it is for a solar application without studying the solar system. It is not a deep scientific work.
Response: MCHS is an important components in solar system and will influence the performance of solar system. Hence, the manuscript is consistent with the theme of present special issue.
2) The validation has to be done with Nusselt number, friction factor and the solar collector efficiency.
Response: A verification with Nusselt number and friction coefficient are done.
3) The PEC parameter and the collector efficiency have to be given. Which is the solar collector efficiency enhancement?
Response: We added some numeric results and percentages according to the suggest of reviewer 3, because such comparison can attract more readers.
4) Merging many references is not permitted.
Response:Revised. Less than 3 references are merged.
5) Using symbols in the abstract is not a good and informative way for presenting your work.
Response: Revised. We added the names of various symbols,
Reviewer 3 Report
The manuscript is much now improved and can be accepted after the following improvements:
The author showed some red color texts, why? The results and discussion do not contain any numeric results and percentages. these comparisons must be added for the readers attraction. Give references of equations 1,2,3,4,6 and 10. Language must be check and a,an and the must be checked as found many cases such as in lines: 42, 96, 98, 135, 143, 145 and many others. remove words of active sentences such as us and we in lines: 16, 99, 223, 234, 263, 273, 275 and 377 line 17, change cooling to heat transfer. the abbreviations in the abstract must be defined before used or removed and use the full word. the ratios at the abstract are not understandable and must be defined. line 34, remove "and so on". line 105, remove "see" and replace with "as shown" the word "burden" is not technical. line 116: put L,W and H next to the length, width and height. line 116: add "respectively" after um. Figure the caption of (a,b and c) must under the figure caption. Figure1, give the thickness of the silicon substrate, Solar cell and the PCB plate. Figure 2, remove (a). line 127, list all assumptions not just for example. give the type of model is it 2D or 3D. line 141 and 142, not useful sentence. Re-write it. line 164 to 166, not useful sentence. Re-write it. Line 170, replace “So”. Specify the boundary conditions such as the inlet temperature, inlet flow rate and outlet pressure. Line 176, what is UDF? Lines 183 to 186, re-write the paragraph. Line 219, replace “go up” by “increase”. Line 223, check using “lie” Line 337, remove “also”.
Author Response
To Reviewer 3
The author showed some red color texts, why?
Response: Red texts indicate what revised.
The results and discussion do not contain any numeric results and percentages. these comparisons must be added for the readers attraction.
Response: Added.
Give references of equations 1,2,3,4,6 and 10.
Response: Eqn. 1-4 from [34], 10 from [42] was marked in original manuscript. Eqn 6 is from [40] and added in revised manuscript.
Language must be check and a,an and the must be checked as found many cases such as in lines: 42, 96, 98, 135, 143, 145 and many others. remove words of active sentences such as us and we in lines: 16, 99, 223, 234, 263, 273, 275 and 377 line 17, change cooling to heat transfer.
Response: Revised.
the abbreviations in the abstract must be defined before used or removed and use the full word.
Response: Revised.
the ratios at the abstract are not understandable and must be defined.
Response: Revised.
line 34, remove "and so on". line 105, remove "see" and replace with "as shown" the word "burden" is not technical. line 116: put L,W and H next to the length, width and height. line 116: add "respectively" after um.
Response: Revised.
Figure the caption of (a,b and c) must under the figure caption. Figure1, give the thickness of the silicon substrate, Solar cell and the PCB plate.
Response: Revised.
Figure 2, remove (a). line 127, list all assumptions not just for example. give the type of model is it 2D or 3D. line 141 and 142, not useful sentence.Re-write it.
Response: Revised.
line 164 to 166, not useful sentence. Re-write it. Line 170, replace “So”.
Response: Revised.
Specify the boundary conditions such as the inlet temperature, inlet flow rate and outlet pressure.
Response: All boundary are specified.
Line 176, what is UDF? Lines 183 to 186, re-write the paragraph. Line 219, replace “go up” by “increase”. Line 223, check using “lie” Line 337, remove “also”.
Response: Revised.
Round 3
Reviewer 1 Report
The authors have not followed my suggestion to study the solar collector as solar collector, Thus I reject this work. However, they could add the PEC values in order to had improved this paper.
Reviewer 3 Report
The paper is much improved and have the following comments:
Remove the word "Ref" in line 80. Line 121: add the word "respectively" after 300um. The bullets numbering in lines 137,139, 141 and 147 are confusing with the section numbering, please change this. Lines 171-174 are need to be rewritten with out the arrows or put it in chart. Paragraph from line 242-251, explains figure 6 but the author refers it to figure 7, please check.Good luck...